# Assessment of Fast-Track Pathway in Hip and Knee Replacement Surgery by Propensity Score Matching on Patient-Reported Outcomes

**DOI:** 10.3390/diagnostics13061189

**Published:** 2023-03-21

**Authors:** Andrea Campagner, Frida Milella, Stefania Guida, Susan Bernareggi, Giuseppe Banfi, Federico Cabitza

**Affiliations:** 1IRCCS Istituto Ortopedico Galeazzi, 20157 Milano, Italyfederico.cabitza@unimib.it (F.C.); 2Faculty of Medicine and Surgery, Università Vita-Salute San Raffaele, 20132 Milano, Italy; 3Dipartimento di Informatica, Sistemistica e Comunicazione, University of Milano–Bicocca, 20126 Milano, Italy

**Keywords:** fast track, propensity score analysis, patient-reported outcome measure, orthopedic, rehabilitation

## Abstract

Total hip (THA) and total knee (TKA) arthroplasty procedures have steadily increased over the past few decades, and their use is expected to grow further, mainly due to an increasing number of elderly patients. Cost-containment strategies, supporting a rapid recovery with a positive functional outcomes, high patient satisfaction, and enhanced patient reported outcomes, are needed. A Fast Track surgical procedure (FT) is a coordinated perioperative approach aimed at expediting early mobilization and recovery following surgery and, accordingly, shortening the length of hospital stay (LOS), convalescence and costs. In this view, rapid rehabilitation surgery optimizes traditional rehabilitation methods by integrating evidence-based practices into the procedure. The aim of the present study was to compare the effectiveness of Fast Track versus Care-as-Usual surgical procedures and pathways (including rehabilitation) on a mid-term patient-reported outcome (PROs), the SF12 (with regard both to Physical and Mental Scores), 3 months after hip or knee replacement surgery, with the use of Propensity score-matching (PSM) analysis to address the issue of the comparability of the groups in a non-randomized study. We were interested in the evaluation of the entire pathways, including the postoperative rehabilitation stage, therefore, we only used early home discharge as a surrogate to differentiate between the Fast Track and Care-as-Usual rehabilitation pathways. Our study shows that the entire Fast Track pathway, which includes the post-operative rehabilitation stage, has a significantly positive impact on physical health-related status (SF12 Physical Scores), as perceived by patients 3 months after hip or knee replacement surgery, as opposed to the standardized program, both in terms of the PROs score and the relative improvements observed, as compared with the minimum clinically important difference. This result encourages additional research into the effects of Fast Track rehabilitation on the entire process of care for patients undergoing hip or knee arthroplasty, focusing only on patient-reported outcomes.

## 1. Introduction

Total hip (THA) and total knee (TKA) arthroplasty procedures have steadily increased over the past few decades [1,2], and their use is expected to grow further [3] as the population ages [1]. Additionally, the arthrosis disorders caused by sports injuries and other physical traumas are contributing factors to the escalating demand for arthroplasties [4].

The increasing number of elderly patients may result in healthcare systems being unable to afford a prolonged hospital stay post operatively or provide intensive rehabilitation services [5]. On the other hand, the expectation of young patients receiving joint replacement is that they will be able to return to normal function, with minimal discomfort, following the procedure [5,6]. Consequently, there is a need for cost-containment strategies, supporting a rapid recovery with a positive functional outcomes, high patient satisfaction, and enhanced patient reported outcomes [4,7].

The Fast Track surgical procedure (FT) is defined as a coordinated perioperative approach [3,8] that strives to lessen the physiologyical and psychological stress associated with surgery, with the aim of expediting early mobilization and recovery following surgery [7] and, accordingly, shortening the length of hospital stay (LOS) [9], decreasing convalescence [10] and lowering costs [11], while maintaining patient safety and without increasing readmission rates [12]. In this view, based on the patient’s condition, rapid rehabilitation surgery optimizes traditional rehabilitation methods, by integrating evidence-based practices into the procedure [13]. Conversely, Care-as-Usual rehabilitation programs result in a longer delay between surgery and rehabilitation activities aimed at functional recovery, leading to delayed discharge, as well as complications associated with a longer hospital stay. Although [14] observed that supervised progressive resistance training was not superior when compared to unsupervised home-based exercise, in the rehabilitation of patients who have had unicompartmental knee arthroplasty (UKA), a recent study by [15] found that patients undergoing UKA, according to a Fast Track and telerehabilitation protocol, had significantly better WOMAC index scores at 2, 15, and 40 days than those undergoing standard surgery and rehabilitation.

Since its introduction in the late 1990s [16], Fast Track has been applied to a variety of surgical disciplines, such as gastrointestinal surgery (e.g., [17,18]), hepatobiliary surgery (e.g., [19]), and, most relevantly, orthopedics [9]. In the orthopedic setting, the effectiveness of the Fast Track surgical procedure as compared to the Care-as-Usual procedure has been widely evaluated, by using standard outcome measures, such as mortality, LOS, or readmission rates (e.g., [20,21,22,23,24,25]).

Nevertheless, few studies appear to emphasize the use of patient-reported outcome measures (PROMs) as opposed to the more conventional standard indicators. For instance, Ref. [26] evaluated the effectiveness of the Fast Track versus Care-as-Usual programs on disease-specific (i.e., pain and global Harris Hip Scores—HHS) and generic post-operative outcomes (i.e., SF-36 physical and mental scores) 12 to 18 months following total hip replacement surgery, without observing significant differences in the SF-36 domain scores, or in the magnitude of the improvement in pain and global HHS. By contrast, Ref. [27] found that both Fast Track and Care-as-Usual rehabilitation programs were effective in terms of health-related outcome measures, such as functional health status (i.e., Functional Independence Measure—FIM) and quality of life (i.e., SF-36), over a 12-month follow-up period, with a faster recovery observed for multi-trauma patients managed with a Fast Track program at 6 months, as compared to 9 months for those assigned to the Care-as-Usual pathway. Ref. [28], on the other hand, found no statistically significant differences in improvement scores on disease-specific (e.g., Knee Injury and Osteoarthritis Outcome Score—KOOS) and generic post-operative outcomes (e.g., EuroQoL-5D) 3 months after unicompartmental knee arthroplasty (UKA) between patients assigned to a Fast Track surgical procedure and those included in an outpatient surgery pathway. On the contrary, Ref. [29] found a statistically significant difference in early improvements, as measured by the WOMAC osteoarthritis index, between the patients in the Fast Track rehabilitation group and those in the standard rehabilitation pathway. These studies are particularly interesting in light of the increasing emphasis toward value-based medicine [30,31,32], as PROMs can more accurately represent the self-perceived health status of patients, as compared with more traditional, hard endpoints. At the same time, one common limitation of the above mentioned studies is the lack of application of techniques that are used to control confounding factors; indeed, some of the same factors that influence the decision to perform a Fast Track surgical procedure also influence the final outcomes, as well as the PROMs scores that are used as a target measure of clinical improvement. To the best of our knowledge [33], only a few studies have used methods for bias correction in the context of analyzing a Fast Track surgical procedure, such as [3], who compared pre-Fast Track and Fast Track patients undergoing elective hip or knee replacement surgery on the Fast Track protocol adherence measure, and [34], who compared the case of Fast Track inpatient pathways with outpatient total joint arthroplasty; both of these works applied propensity score-matching (PSM). In contrast, except for [29] (who applied PSM to analyze the impact of Fast Track and tele-rehabilitation, using the WOMAC PROM score as a target for improvement), none of the reviewed studies that have compared the effectiveness of the Fast Track program versus the traditional ones, in terms of PROMs, have used PSM (or similar bias-correction methods) to ensure the homogeneity of the comparison populations (Fast Track vs. Care-as-Usual), regarding confounding variables that may affect both the assigned treatment and the outcome of interest.

The aim of the present study was to compare the effectiveness of Fast Track versus Care-as-Usual surgical procedures on a mid-term patient-reported outcome, the SF12 (with regard both to Physical and Mental Scores), 3 months after hip or knee replacement surgery, with the use of propensity score-matching analysis to address the issue of the comparability of the groups in a non-randomized study. We were interested in the evaluation of the entire pathways, including the postoperative rehabilitation stage, therefore, we only used early home discharge as a surrogate, in order to differentiate between Fast Track and conventional rehabilitation pathways. More in detail, we evaluated the following research questions and hypotheses:(RQ1): Does the Fast Track pathway (including rehabilitation) have a positive impact on health status (as measured by the SF-12 Physical and Mental scores), compared with the Care-as-Usual one?(RQ2): Does the Fast Track pathway lead to a better improvement in health status (measured with respect to the pre-operative stage), compared with the Care-as-Usual one?

## 2. Methods

To the aim of evaluating the effectiveness of the Fast Track process, as compared with Care-as-Usual, we performed a statistical analysis based on a inferential statistics approach. The analysis was performed on a dataset extracted from the electronic health records (EHRs) of the IRCCS Ospedale Galeazzi - Sant’Ambrogio (henceforth OGSA), one of the major orthopedic research hospitals in Italy, which encompasses a total of 1600 records, for as many single patients. The study was designed as a retrospective cohort study; in the following description of the methods, we comply with the STROBE checklist for cohort studies [35].

### 2.1. Study Design and Cohort Description

The dataset included patients admitted to OGSA for elective hip or knee replacement surgery between January 2013 and February 2022. The data for the patients encompassed both demographic information (age; sex; provenance, i.e., lives in the same region as the hospital, our outside the region) and clinical information (diagnosis; ASA class; BMI; length of stay; length of surgery; septic status, i.e., whether patients developed a septic infection as a consequence of the surgery), as well as information drawn from PROMs (SF12 questionnaire, Mental Score and Physical Score [36]), and this information was collected both at pre-operative times and 3 months after surgery. In particular, the 3-month SF12 scores were considered to be the target criteria for analyzing the benefit of Fast Track. Data features and their characteristics are reported in Table 1.

Since there were records with missing values, we removed all patients for which at least one of the considered features was missing. Upon consultation with experts from the OGSA, it was established that patients were admitted to the Fast Track based on a preoperative, doctor-filled questionnaire, aimed at evaluating the necessity of hospital stay after surgery (Care-as-Usual). In particular, the following criteria were established as being sufficient for exclusion from the Fast Track: being of an age older than 80 years, presence of a comorbidity with ASA class III, complexity of the pathology or of the surgical procedure, septic infection, secondary or major revision intervention. To avoid bias in the analysis, all the patients satisfying at least one of these criteria were excluded from further analysis.

After missing data removal and patient exclusion, the considered cohort for statistical comparison encompassed 1053 patients, of which 694 patients were assigned to the Fast Track.

Regarding the differences in the rehabilitation pathway, between the Fast Track and Care-as-Usual procedures, patients admitted to the Fast Track are subject to early mobilization (between 4 and 6 h after surgery) and, after agreement with anesthetist, the patient is upright and allowed to walk in the first day after surgery. For both Fast Track and Care-as-Usual patients, rehabilitation consists of two 30 min sessions of physiotherapy each day. Since the main difference in the two pathways were the early mobilization and early beginning of rehabilitation, we did not consider other factors (e.g., number of rehabilitation sessions) as potential confounders.

### 2.2. Statistical Methods: Propensity Score Matching

Since most of the considered covariates influenced both the assignment to the Fast Track and the outcome (see Figure 1, for a graphical representation of the dependence relationships between covariates), we performed a statistical analysis based on *propensity score-matching* (PSM) [37], so as to allow for the unbiased estimation of the causal effects of the assignment to the Fast Track on the two considered outcomes. PSM is a statistical method used to remove confounding bias from observational cohorts where randomization is unfeasible [38], offering advantages over more traditional regression methods [39]. In particular, to implement PSM, we corrected for all covariates influencing both assignment to the Fast Track and outcomes. In addition, we also corrected for BMI, sex and main location (knee or hip), in order to enforce a strong comparability requirement on the considered populations.

Propensity scores were computed by means of a logistic regression model, which was trained on the 1053 non-excluded patients and all considered covariates (see Figure 1); assignment to the Fast Track vs. Care-as-Usual was considered as the target dependent variable. Before performing matching, we evaluated a set of standard statistical checks to ensure that the propensity score model was well-specified; in particular, we performed the following statistical tests [40]:In each decile of the distribution of propensity scores, the distribution of propensity scores for the Fast Track and Care-as-Usual populations should be the same;In each decile of the distribution of propensity scores, the distribution of each covariate in the Fast Track and Care-as-Usual populations should be the same;

For all the above-mentioned tests, the non-parametric Mann–Whitney U procedure was applied. Matching was performed conditionally on the failure to reject the null hypothesis of no difference between populations, for all the above-mentioned statistical tests.

PSM was performed by means of the *optimal matching* procedure [41], by which each Fast Track patient was matched to a different Care-as-Usual patient, so as to minimize the total distance in propensity scores between the two populations. We matched 359 patients for each of the two populations we selected, for a total of 718 patients. Statistical analysis of the outcomes for the two populations was performed, and was conditional on the following statistical checks of the results of the matching procedure [42]:Quantile–quantile plot qualitative analysis: for each of the corrected-for covariates, the post-match bi-variate distribution across Fast Track and Care-as-Usual patients should be closer to the diagional of the quantile–quantile plot than the same distribution for the pre-matched samples;Standardized mean differences (SMD) analysis: all corrected-for covariates should have a standardized mean difference smaller than 0.1 (which corresponds to a negligible effect size in the difference between the two populations [43])Variance ratio (VR) analysis: the ratio of the Fast Track and non-Fast Track sample variances should be smaller than 2.

### 2.3. Statistical Methods: Analysis

After PSM, statistical analysis was performed by comparing the outcome values (SF12 Physical Score and SF12 Mental Score, both at 3 months from surgery) between the two populations, using the Mann–Whitney U statistical test with the Rank Biserial Correlation (RBC) as a measure of effect size. We also compared the proportion of patients in the two populations whose target outcome improved; improvement was defined as having a difference between the 3 months and pre-operative scores that was greater than the *minimum clinically important difference* (MCID), as defined through a distribution-based approach [44]. Comparison among proportions of improvement was performed using McNemar $\chi^{2}$ test, with Cohen *d* as a measure of effect size; the effect size was also used to compute the *number needed to treat* (NNT) (i.e., the average number of patients who need to be assigned to the Fast Track to prevent one additional negative outcome), as an alternative measure of effect size. The NNT was computed by means of the formula $NNT=\frac{1}{2\Phi (\frac{d}{\sqrt{2}})-1}$ [45], where *d* is the Cohen *d* effect size and Φ is the cumulative distribution function of a standard normal distribution.

## 3. Results

The distribution of propensity scores in the pre-matched distributions is reported in Figure 2a; the two distributions were significantly different (Mann–Whitney U test: *p* < 0.001).

The pre-match test on propensity scores’ distributions failed to reject the null hypothesis of equal distribution across the decile strata (adjusted *p*-values: 1, 1, 1, 1, 1, 1, 0.056, 1, 0.782, 0.782). Similarly, the pre-match test on the distribution of covariates failed to reject the null hypothesis of equal distribution across the decile strata (see Figure 3). Hence, matching was performed.

The distribution of propensity scores in the matched sub-population is reported in Figure 2b; the two distributions were not statistically significantly different (Mann–Whitney U test, *p*: 0.274). The quantile–quantile plots for the continuous variables are reported in Figure 4a–d. The SMD values for the pre-matching and post-matched populations, for both continuous and categorical covariates, are reported in Figure 5a; in the pre-matched populations, the SMDs were all larger than 0.1, expect for the out of region covariate. In contrast, in the post-matched populations, all SMDs were smaller than 0.1, and all covariates except for the SF12 Mental Score and SF12 Physical Score had a SMD smaller than 0.05. The VRs, for continuous covariates, are reported in Figure 5b; all VR values were smaller than two, for both the pre-matched and post-matched populations.

Since all the post-matching statistical checks were successful, we compared the distributions of outcomes in the Fast Track and Care-as-Usual populations, by considering the SF12 Physical and Mental Scores measured at 3 months after surgery. The comparison between the raw outcome scores is represented graphically in Figure 6a,b. For the SF12 Physical Score, the difference between the two populations was statistically significant (Mann–Whitney U test, *p*: 0.002), but was associated with a small effect size (RBC: 0.13); patients in the Fast Track population reported, on average, a higher score than patients in Care-as-Usual population (43.46 vs. 41.38). In contrast, for the SF12 Mental Score, the difference between the two populations was not statistically significant (Mann–Whitney U test, *p*: 0.646) and was associated with a negligible effect size (RBC: 0.02). The comparison between the rates of improved patients for the two outcomes is represented graphically in Figure 7a,b. For the SF12 Physical Score, the difference in the proportion of improved patients between the two populations was statistically significant (McNemar $\chi^{2}$ test, *p*: 0.026) but was associated with a small effect size (Cohen d: 0.16): the Fast Track population had a larger proportion of patients whose outcome improved than the Care-as-Usual population (62.9% vs. 54.9%), and the NNT for Fast Track was 10.80. In contrast, for the SF12 Mental Score, the difference in the proportion of improved patients between the two populations was not statistically significant (McNemar $\chi^{2}$ test, *p*: 0.415) and was associated with a negligible effect size (Cohen d: 0.06); however, the Fast Track population had a smaller proportion of patients whose outcomes improved compared with the Care-as-Usual population (23.1% vs. 25.9%), and the NNT for Care-as-Usual was 27.4.

## 4. Discussion

In this article, we studied the effectiveness of Fast Track surgical procedures, compared with Care-as-Usual, by adopting a patient-centered perspective and measuring the impact of these procedures on the PROMs, indicators reported by patients themselves. Even though few previous studies have focused on such a perspective [33], we believe this latter to be not only complementary to approaches based on standard hard endpoints, but also of extreme practical interest, due to the role played by PROMs in value-based healthcare practice [30,31,32].

More in detail, we compared the effectiveness of the Fast Track pathway (which also encompasses rehabilitation), as compared with the standardized process, on the SF12 Physical and Mental Scores, as the SF12 is one of the most widely used instruments for assessing self-reported HRQoL [46]. We focused on the impact of Fast Track protocol intervention on mid-term patient-reported outcomes, that is, 3 months after surgery, because we were interested in comparing the effectiveness of the entire pathways, including the postoperative rehabilitation stage. We only used early home discharge as a surrogate to differentiate between the Fast Track and conventional rehabilitation; thus, in this study, all the patients included in the Fast Track protocol following hip or knee replacement surgery were discharged to their homes. A propensity score-matching approach (based on logistic regression and optimal matching) was used to account for systematic differences in baseline characteristics between treated and untreated subjects, as well as to obtain unbiased estimations of treatment effects on outcomes. We believe this to be a particularly strong design choice of our study, since only one of the reviewed previous studies measuring the impact of these procedures on the PROMs used PSM (or similar bias correction methods) to address the issue of the comparability of the groups, in a non-randomized study.

Our findings show that the Fast Track procedure had a positive impact on the perceived physical functioning status 3 months after the operation, because the reported physical enhancement differed to a statistically significant extent between the study groups. Patients who adhered to the Fast Track pathway showed higher scores for the SF12 physical domain than those reported, on average, by patients managed with the traditional pathway. However, the associated effect size indicates that the magnitude of the improvement provided by the Fast Track program is relatively small. In contrast, our study demonstrates that no difference was detectable between these procedures when patients reported their own perceptions of their mental health 3 months after surgery. This may lead one to consider that the effectiveness of the Fast Track procedure is only relevant to the physical health-related status, while the reported benefit on mental status is negligible 3 months after the operation. Interestingly, when we compared these procedures by the absolute improvement (MCID), we observed that patients in the Fast Track population had a significantly larger rate of improvements in terms of physical scores, as compared to the Care-as-Usual cohort. On the contrary, we found no statistically significant difference between the Care-as-Usual population and the Fast Track one when considering mental scores (though, on average, the Care-as-Usual population had a higher proportion of patients whose outcomes improved compared to the Fast Track). Notwithstanding, the NNT estimated for the Care-as-Usual pathway, in terms of mental scores, shows that the traditional pathway is significantly less effective, needing around 28 patients to be assigned to standard treatment for only one of them to exhibit minimum clinically important difference improvement. In contrast, the NNT estimated for the Fast Track pathway in terms of physical scores confirms the prior evidence that the Fast Track procedure is more effective than the Care-as-Usual pathway, requiring only the assignment of 11 patients to the Fast Track procedure for one of them to perceive a minimum clinically important difference improvement of their physical status 3 months after surgery. It is worth noting that quality of life is a broad concept that is only partly affected by clinical symptoms [47]. Therefore, further studies should be conducted, in order to confirm the results obtained. On the other hand, the minimum clinically important difference (MCID) is defined as the lowest change in PROM scores that patients perceive as meaningful [48]; hence, future research should use MCID as target outcome as it allows for comparing differences in the effectiveness of these procedures on the PROMs, on the basis of the patient-centered prospective. Furthermore, most of the studies we found that used PSM as a methodology for bias correction have shown that the use of the Fast Track surgical procedure is effective in terms of clinical outcomes. Ref. [3] observed improvements in pain management, a reduction in hemoglobin loss and the frequency of transfusions and blood re-infusions, and a reduction in the adverse outcomes and complications throughout the intermediate and long-term, in the cases of elective hip and knee replacement. Ref. [34] demonstrated that 30 day readmission rates were not greater after outpatient surgery when compared to Fast Track inpatient procedures, even though complications and re-operations significantly increased after total joint arthroplasty. In contrast, Ref. [49] showed that unicompartmental knee arthroplasty (UKA) was related to greater functional scores than total knee arthroplasty (TKA), without a substantial increase in the number of revisions. Even though this latter study is outside of the Fast Track setting, and none of the above-mentioned studies use PSM to compare the procedures on the PROMs, the studies confirm the increasing adoption of bias-correcting techniques, such as PSM, to eliminate confounders in the comparison of cohorts of patients in orthopedic settings. As we mentioned in the introduction, only the recent study by De Berardinis et al. [15], applied PSM to analyze the impact of the Fast Track on PROMs: in particular, the authors found that patients undergoing UKA, according to a fast-track and telerehabilitation protocol, had significantly better WOMAC index scores at 2, 15, and 40 days than those undergoing standard surgery and rehabilitation, using the PSM methods. When compared with this previous study, our results are more general in terms of the considered procedures (in that they apply to both hip- and knee-related procedures, rather than only knee-related ones), and also we consider the SF-12 scores, rather than WOMAC, which, as mentioned before, is one of the most common tools used to measure HRQoL. These results, combined with those reported in the articles mentioned above, lead to strengthening the importance of such an approach, in terms of studies aimed at investigating the effectiveness between Fast Track and Care-as-Usual procedures on PROMs.

This study has some limitations. First, we only used early home discharge as a proxy to distinguish between the Fast Track and the rehabilitation program. Future research should also provide a reference standard for the post-discharge rehabilitation stage, in addition to a characterization of the two rehabilitation modalities, before discharge from the orthopedic hospital. This would allow us to validate the beneficial impact of a Fast Track approach to rehabilitation on patients’ perceptions of their health status at follow-ups. Second, we measured the effectiveness of the Fast Track procedure, as compared to the Care-as-Usual pathway, on generic patient-reported outcome measures, that is, the SF12 Physical and Mental Scores. Future research should integrate the analysis by adopting disease-specific PROMs or alternative quality of life instruments, in order to confirm the enhancement perceived by Fast Track patients of their physical status 3 months after operation. Third, the number of rehabilitation sessions performed by the patient in the three post-operative months was not considered. Though, as mentioned in the methods, the Fast Track and Care-as-Usual pathways did not differ in terms of the number and length of the rehabilitation sessions, it is, however, conceivable that patients admitted to different pathways would exhibit different adherence levels to the rehabilitation plan. In fact, a better result in PROMs could be attributable to greater adherence to the rehabilitation plan or simply to the opportunity of taking advantage of a home rehabilitation service, rather than an outpatient one. This limitation is especially relevant, as in Italy there is a large inter-regional variability regarding access to rehabilitation care through the national health system or through private practice; obviously the transportability of our results depends on the specifics of patients’ access to rehabilitation care. Nonetheless, as the OGSA (i.e., the data collection setting) is one of the major orthopedic hospitals in Italy, we believe our results to be generalizable to other similar settings, especially as we relied on matching techniques in our statistical analysis, which aim at making the groups compared homogeneous. Future work should focus on taking into account the adherence of patients to the rehabilitation plan, and its impact on PROMs.

## 5. Conclusions

Our study shows that the entire Fast Track pathway, which incorporates the post-operative rehabilitation stage, has a positive impact on physical health-related status (SF-12 physical scores) perceived by patients 3 months after hip or knee replacement surgery, as opposed to the standardized program. This result encourages additional research into the effects of Fast Track rehabilitation on the entire process of care for patients undergoing hip or knee arthroplasty, focusing only on patient-reported outcomes. Therefore, the study contributes to measuring the impact that innovative procedures for surgical treatment, such as the Fast Track surgical procedure in the context of the current study, may have on patients’ quality of life (by employing a proxy of their own perceived health status, the scores on PROMs), allowing for further research into the potential organizational outcomes that could arise from a pathway that employs multidisciplinary teams and, requires coordinated intervention across all phases of perioperative care [3].

## Figures and Tables

**Figure 1 diagnostics-13-01189-f001:**
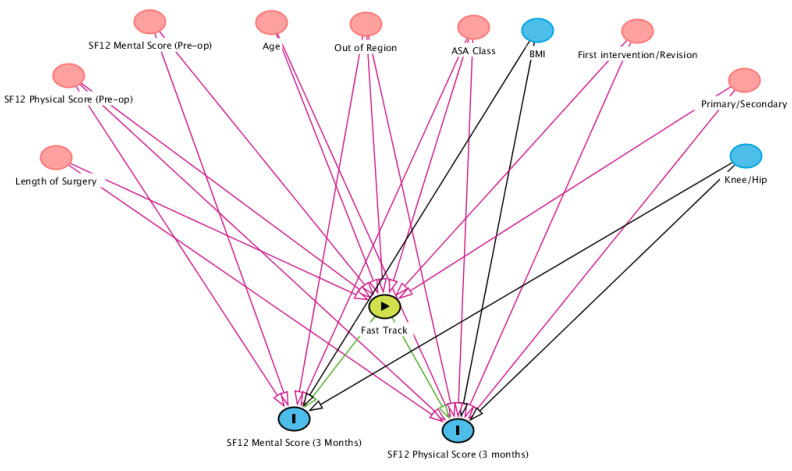
Directed acyclic graph representing the relationships between the considered covariates, the outcomes (SF12 Physical and Mental scores, 3 months after surgery) and intervention (assignment to Fast Track). Green paths denote the direct causal effect of the intervention on the outcome, while violet arrows denote biasing paths; red nodes, in particular, represent confounders (covariates that influence both assignment to treatment and the outcome). The minimal conditioning set was the following: length of Surgery, SF12 Physical Score (Pre-op), SF12 Mental Score (Pre-op), age, out of region, ASA class, first intervention/revision, primary/secondary.

**Figure 2 diagnostics-13-01189-f002:**
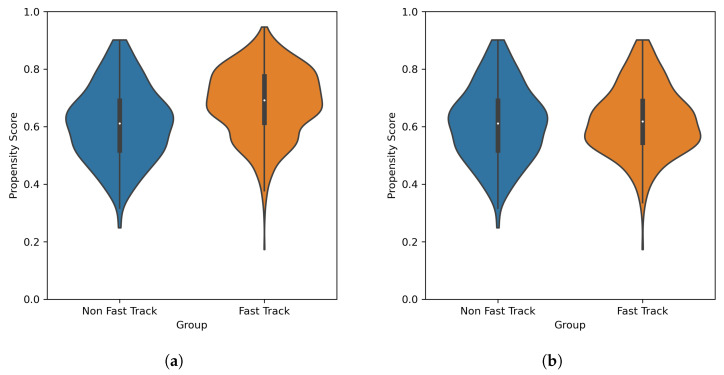
Violinplot of the distribution of propensity scores in the pre-matched (**a**) and post-matched (**b**) populations.

**Figure 3 diagnostics-13-01189-f003:**
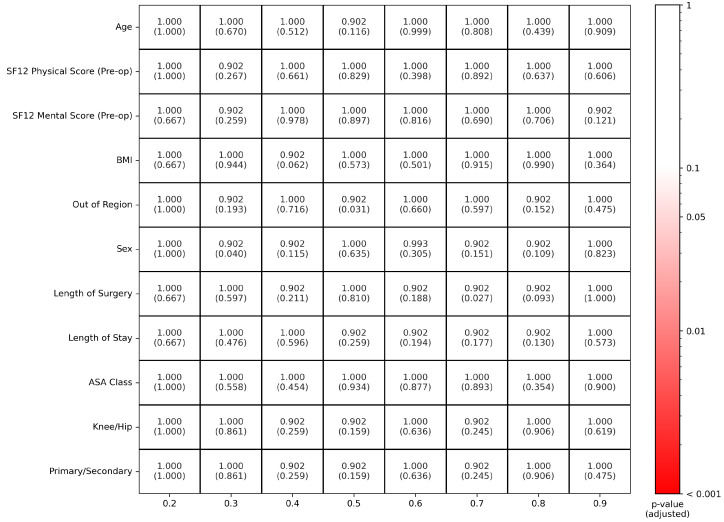
Decile comparison tests for all the corrected-for covariates.

**Figure 4 diagnostics-13-01189-f004:**
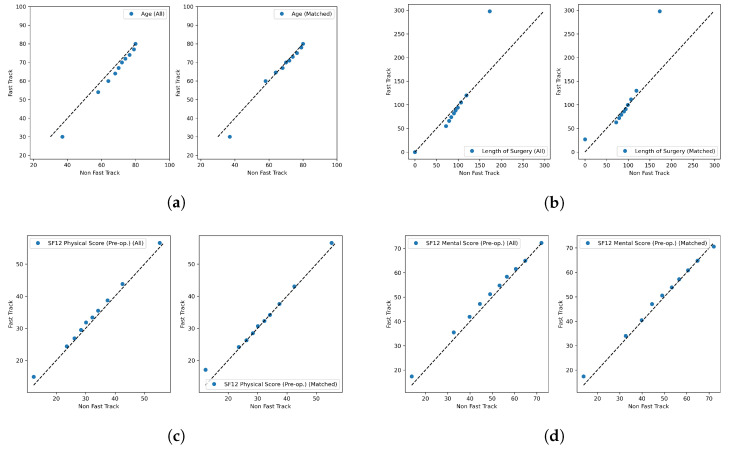
Quantile–quantile plots of the corrected-for continuous covariates: Age (**a**), Length of Surgery (**b**), SF12 Physical Score pre-op (**c**), SF12 Mental Score pre-op (**d**).

**Figure 5 diagnostics-13-01189-f005:**
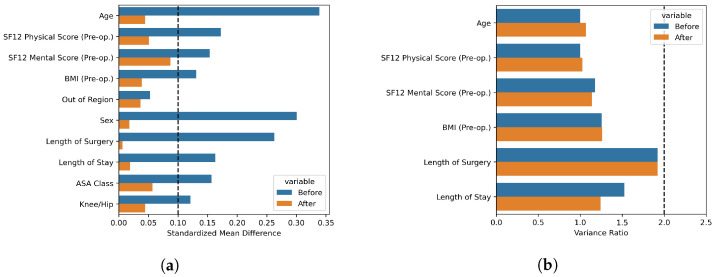
Boxplots for the standardized mean differences (**a**) and variance ratios (**b**) of the corrected-for covariates.

**Figure 6 diagnostics-13-01189-f006:**
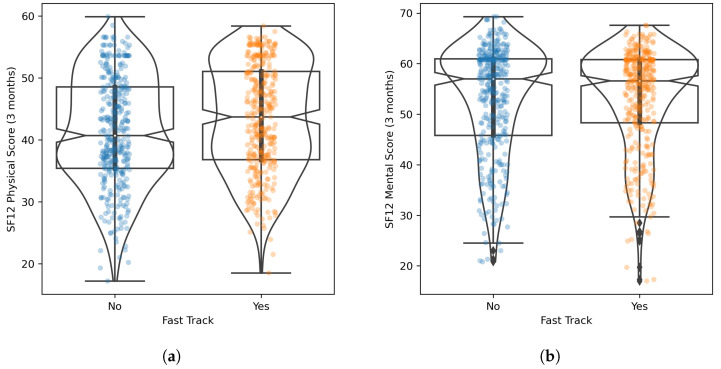
Boxplots, violinplots and stripplots for the distribution of the SF12 Physical Score (**a**) and SF12 Mental Score (**b**) for the two populations, at 3 months after surgery.

**Figure 7 diagnostics-13-01189-f007:**
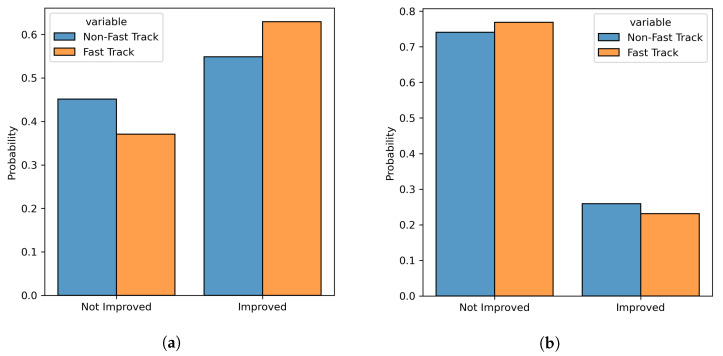
Boxplots for the proportion of patients who improved in terms of SF12 Physical Score (**a**) and SF12 Mental Score (**b**) for the two populations, at 3 months after surgery.

**Table 1 diagnostics-13-01189-t001:** Descriptive statistics for the available data features.

Feature	Mean	Std. Dev.	Missing Rate
Age (years)	68.72	10.96	0.00
SF12 Physical Score (Pre-op)	32.10	7.70	0.00
SF12 Mental Score (Pre-op)	49.69	12.54	0.00
BMI	27.39	4.73	0.01
SF12 Physical Score (3 months)	42.07	9.32	0.00
SF12 Mental Score (3 months)	52.56	10.71	0.00
Length of Surgery (minutes)	93.98	34.48	0.02
Length of Stay (days)	4.01	1.95	0.02
Feature	Categories		Missing Rate
Fast Track	Yes (66%), No (34%)		0.00
Out of Region	Yes (18%), No (82%)		0.00
Sex	Male (59%), Female (41%)		0.00
ASA Class	1 (11%), 2 (82%), 3 (7%)		0.00
Knee/Hip	Knee (55%), Hip (45%)		0.00
Primary/Secondary	Primary (84%), Secondary (16%)		0.00
First intervention/Revision	First intervention (91%), Revision (9%)		0.00

## Data Availability

All data used for the analysis are available on the Zenodo platform, at https://zenodo.org/record/7750128#.ZBgsifbMK3A accessed on 16 March 2023.

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
