# Peer review of "Assessment of Fast-Track Pathway in Hip and Knee Replacement Surgery by Propensity Score Matching on Patient-Reported Outcomes"

_diagnostics, 2023, doi:10.3390/diagnostics13061189_

Round 1

Reviewer 1 Report

Dear Authors,

the manuscript is very interesting and well-written. I have only few minor issues that need to be solved before publication:

- The Material section should be divided into few parts like participants, assessment methods, classification procedures etc. to make it easier to understand.

- Figure 3 is too small and therefore not readable.

- The Discussion section must be improved - the results obtained in this study need to be compared to other studies, even if there are only few. It would be worthful to analyse other studies, that used the same assessment scores, even if they did not assessed fast-track protocols. Whis can lead to some additional conclusions.

Author Response

A comparison between the previous version of the manuscript and the current one, for the reviewer'’s convenience, can be seen at: 

https://draftable.com/compare/ukIIcAqPtrJm

Comment

Dear Authors,

the manuscript is very interesting and well-written. I have only few minor issues that need to be solved before publication:

Response

We thank the reviewer for their positive assessment of the manuscript

Comment

- The Material section should be divided into few parts like participants, assessment methods, classification procedures etc. to make it easier to understand.

Response

We have divided the Methods section into three sub-sections: Study Design and Cohort Description, Statistical Methods (Propensity Score Matching) and Statistical Methods (Analysis)

Comment

- Figure 3 is too small and therefore not readable.

Response

We have increased the size of Figure 3

Comment

- The Discussion section must be improved - the results obtained in this study need to be compared to other studies, even if there are only few. It would be worthful to analyse other studies, that used the same assessment scores, even if they did not assessed fast-track protocols. Whis can lead to some additional conclusions.

Response

We thank the reviewer for their suggestion: we expanded the Discussion section adding some pointers to the literature and some comparisons with related studies. We remark here that, as also we report in the article, up to our knowledge there is only one other study that applied PSM to analyze the impact of Fast Track on PROMs: we discuss about this article and the differences with respect to our work in the Discussion section.

Reviewer 2 Report

I thank the authors for carrying out this interesting study on such a frequent topic as hip and knee arthroplasty and the hospital approach methods: fast track and usual care.

I would like some clarifications regarding the different sections:

Introduction: I don't see the hypothesis expressed after the aim. I ask you to implement this part.

I ask you to address the aspect of rehabilitation in the introduction: in fact, a patient who performs Fast-Track will carry out more rehabilitation in a home or outpatient setting, while the patient who remains hospitalized will benefit more from hospital rehabilitation. In addition to the impact of costs, are there differences described in the literature in terms of efficacy regarding rehabilitation in hospital compared to a home or outpatient rehabilitation setting? Furthermore, it could be interesting to report if there are differences in this patient population (hip and knee replacement) in terms of PROMs and general outcomes after a supervised (with the physiotherapist) or unsupervised rehabilitation (brochures or online guides).

Methods: I don't find that the type of study carried out has been correctly expressed, I ask you to express it better. In the light of the type of study carried out, I ask you to follow the appropriate guidelines: is it a retrospective observational study? If so, I ask you to comply with the STROBE guidelines for observational studies.

Discussion and conclusions: what is the clinical impact of this study? what repercussions does it have at the level of the patient, of the healthcare staff? I ask you to better express this element. In my opinion, it should be included within the limits of the study that the number of rehabilitation sessions performed by the patient in the three post-operative months was not considered: in fact, a better result in the PROMs could be attributable to greater adherence to the rehabilitation plan or simply to the possibility of take advantage of a home rehabilitation service rather than an outpatient one. In Italy there is a large inter-regional variability regarding access to rehabilitation care through the national health system or through private practice. Please disclose this element for the purpose of a better interpretation of the results.

Best regards

Author Response

A comparison between the previous version of the manuscript and the current one, for the reviewer’s convenience, can be seen at: 

https://draftable.com/compare/ukIIcAqPtrJm

Reviewer 2

Comment

I thank the authors for carrying out this interesting study on such a frequent topic as hip and knee arthroplasty and the hospital approach methods: fast track and usual care.

I would like some clarifications regarding the different sections

Response

We thank the reviewer for the positive assessment of the manuscript and their useful comments which helped improving the article

Comment

Introduction: I don't see the hypothesis expressed after the aim. I ask you to implement this part.

Response

We have explicitly added the research questions after the aim

Comment

I ask you to address the aspect of rehabilitation in the introduction: in fact, a patient who performs Fast-Track will carry out more rehabilitation in a home or outpatient setting, while the patient who remains hospitalized will benefit more from hospital rehabilitation. In addition to the impact of costs, are there differences described in the literature in terms of efficacy regarding rehabilitation in hospital compared to a home or outpatient rehabilitation setting? Furthermore, it could be interesting to report if there are differences in this patient population (hip and knee replacement) in terms of PROMs and general outcomes after a supervised (with the physiotherapist) or unsupervised rehabilitation (brochures or online guides).

Response

We thank the reviewer for suggesting we discuss this important point. We note that in the article we consider surgical Fast Track: In regard to differences in the rehabilitation pathway, between the Fast Track and Care-as-Usual surgical procedures, patients admitted to Fast Track are subject to early mobilization (between 4 and 6 hours after surgery) and, after agreement with anesthetist, the patient is upright and allowed to walk already in the first day after surgery. For both Fast Track and Care-as-Usual patients, rehabilitation consists in two 30 minutes sessions of physiotherapy each day. Thus, the main difference in the two pathways were the early mobilization and early beginning of rehabilitation, and not the number of inpatient/outpatient sessions. In regard to differences in terms of efficacy regarding rehabilitation in hospital compared to a home or outpatient rehabilitation setting, we added some references in the article. In regard to differences in terms of PROMS after supervised or unsupervised rehabilitation, though we believe this question to be interesting, we plan to investigate this (also in terms of related work) in future work, as it is not directly related to our aim in this article.

Comment

Methods: I don't find that the type of study carried out has been correctly expressed, I ask you to express it better. In the light of the type of study carried out, I ask you to follow the appropriate guidelines: is it a retrospective observational study? If so, I ask you to comply with the STROBE guidelines for observational studies.

Response

The reviewer is correct, the study is a retrospective observational (cohort) study. We reviewed the article in order to comply with the STROBE guidelines for this type of study.

Comment

Discussion and conclusions: what is the clinical impact of this study? what repercussions does it have at the level of the patient, of the healthcare staff? I ask you to better express this element. In my opinion, it should be included within the limits of the study that the number of rehabilitation sessions performed by the patient in the three post-operative months was not considered: in fact, a better result in the PROMs could be attributable to greater adherence to the rehabilitation plan or simply to the possibility of take advantage of a home rehabilitation service rather than an outpatient one. In Italy there is a large inter-regional variability regarding access to rehabilitation care through the national health system or through private practice. Please disclose this element for the purpose of a better interpretation of the results

Response

In regard to the clinical impact of the study, our results show that given two patients with similar pre-surgical conditions, Fast Track surgical procedures will lead to a better health status than Care-as-Usual ones. Thus, perioperative Fast Track (and associated rehabilitation pathway) can be of great benefit to patients whose health condition allows it: patients undergoing this pathway will have a more rapid and greater improvement in physical condition as compared to whether these same patients would be treated through the Care-as-Usual procedure. Obviously, Fast Track can also have a positive impact in terms of costs and occupancy of resources. In regard to the limitation mentioned by the reviewer, we agree: though the rehabilitation procedures for Fast Track and Care-as-Usual are equal in the number of rehabilitation sessions, it is possible that patients admitted to Fast Track would exhibit a better adherence to the rehabilitation plan. For this reason, we explicitly added this limitation to the article, and we plan to investigate this issue in further work. In regard to the aspect of differences in rehabilitation care with respect to different regions or national/private institutes, we explicitly stated that patients were collected at the IRCCS Ospedale Galeazzi - Sant’Ambrogio (Milan, Italy). Though, obviously the transportability of our results depends on the specifics of  the access to rehabilitation care for the patients involved, we believe our results to be generalizable to other similar settings, especially as we relied on matching techniques in our statistical analysis, which aim at making the groups compared homogeneous.
